# Object-centric 3D Motion Field
# for Robot Learning from Human Videos

**Zhao-Heng Yin**[1]     **Sherry Yang**[1,2]     **Pieter Abbeel**[1]
[1]BAIR, UC Berkeley EECS
[2]Google DeepMind

`https://zhaohengyin.github.io/3DMF`

## Abstract

Learning robot control policies from human videos is a promising direction for scaling up robot learning. However, how to extract action knowledge (or action representations) from videos for policy learning remains a key challenge. Existing action representations such as video frames, pixelflow, and pointcloud flow have inherent limitations such as modeling complexity or loss of information. In this paper, we propose to use object-centric 3D motion field to represent actions for robot learning from human videos, and present a novel framework for extracting this representation from videos for zero-shot control. We introduce two novel components in its implementation. First, a novel training pipeline for training a "denoising" 3D motion field estimator to *extract* fine object 3D motions from human videos with noisy depth robustly. Second, a dense object-centric 3D motion field *prediction architecture* that favors both cross-embodiment transfer and policy generalization to background. We evaluate the system in real world setups. Experiments show that our method reduces 3D motion estimation error by over 50% compared to the latest method, achieve $55\%$ average success rate in diverse tasks where prior approaches fail ($\lesssim 10\%$), and can even acquire fine-grained manipulation skills like insertion.

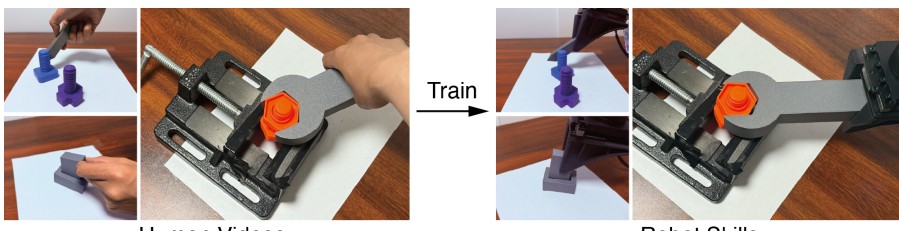

Human Videos        Robot Skills

Figure 1: We propose a novel framework for robot learning from human demonstration videos *without* relying on any robot-collected data. Our approach learns to control robots by extracting and modeling 3D object motion fields from RGBD human videos.

## 1 Introduction

Data is the primary bottleneck in robot learning – collecting large-scale high quality robotic data in real world at scale for training control policies is not only expensive but also mentally challenging for humans in complex tasks [1, 55]. Recently, human-object interaction videos stand out as a particularly promising avenue to overcome this challenge. These videos are not only scalable—given the vast

---

Correspondence to: `zhaohengyin@cs.berkeley.edu`

39th Conference on Neural Information Processing Systems (NeurIPS 2025).

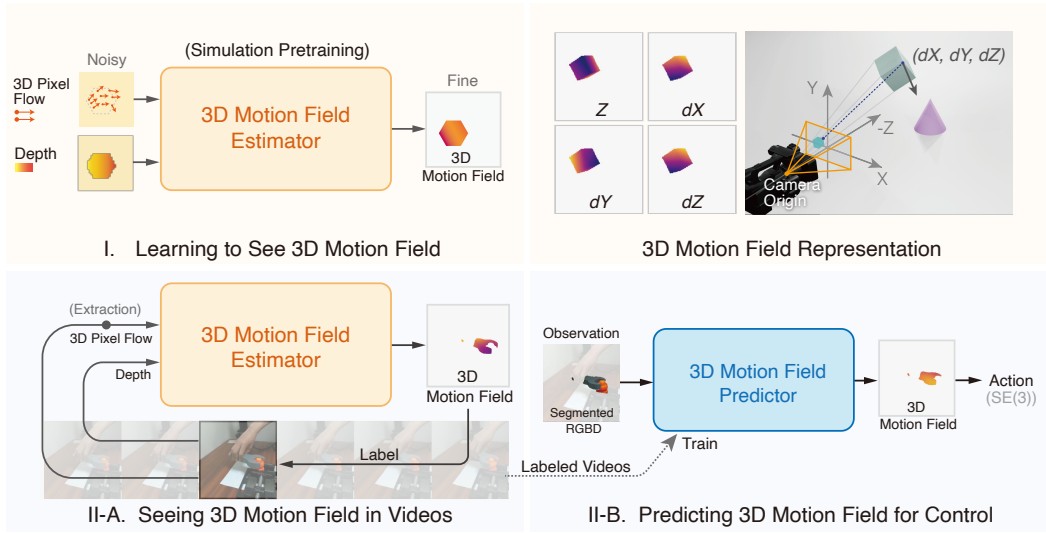

Figure 2: Overview of proposed learning framework. We first pretrain a 3D motion field estimator in simulation (Phase I) and use it to estimate the 3D motion field in *noisy* RGBD human videos (Phase II-A). Then, we train policies to predict the estimated 3D motion field and use them to control robots in a zero-shot manner (Phase II-B). *Unlike* existing 3D tracking works that assume depth as a groundtruth reference, we recover accurate 3D object motion from noisy depth.

amount of footage available from internet or wearable device recordings — but they also capture rich, naturalistic demonstrations of complex tasks. Moreover, collecting data through human hands is inherently easier, cheaper, and more intuitive than through robotic teleoperation [44, 8, 39].

Due to this data collection challenge, many works look into the feasibility of using real-world action-free videos for robot learning. Some recent works, such as UniPi [9] and UniSim [54], directly apply video prediction for control, which essentially view future video frames as action representations and use inverse dynamics model to translate the predicted future frame to actions. Although this line of work achieved some preliminary success, video frames turn out to be an overly noisy, redundant action representation, which not only unnecessarily complicates training and inference but also makes policy non-robust due to blurry details in videos. Therefore, recent works also explored more compact action representations extracted from videos, such as pixel-flow [49, 52, 3], point-cloud flow [59], and SE(3) pose transformation [15]. Nevertheless, each of these representations suffers from certain drawbacks. Pixel-flow is a 2D representation and drops the important 3D movement information for recovering actions. Point-cloud 3D flow is noisy and cannot represent motion accurately. SE(3) pose extraction relies on object 3D models and limits themselves to rigid bodies. We refer readers to the related works (Section 6) for a complete discussion and comparisons. In conclusion, so far, it remains unclear what serves as a good and feasible action representation for video-based robot learning.

In this work, we propose to use an object-centric 3D motion field representation as the action representation for control. Specifically, it is a dense position and motion field over the object pixels in the input image, representing how each observable point on an object should move in each task (Figure 2). This action representation has many advantages compared to previous works: 1. It preserves minimal sufficient 3D information for robot control; 2. It is a image-based representation and allows the use of well-studied powerful image generative models; 3. It is object-centric and embodiment-agnostic, simplifying cross embodiment transfer; 4. Its extraction fully depends on RGBD video and does not assume additional information like object 3D models.

While it may sound appealing, extracting such information from video turns out to be challenging. Although it is relatively simple to extract accurate 2D pixel flow movement part using the latest tracking model, the depth channel is full of noisy measurement values (e.g. missing or wrong values) and directly using the raw depth values with for computing 3D motion will only result in inaccurate motion. Our key insight is that we can build a "denoising" 3D motion field estimator to reconstruct 3D position and motion flow robustly from the noisy depth measurement with simulation data, as depth noise has some easy-to-simulate characteristics. Since this task is fully geometrical and does

not involve complex RGB textures, this simulation-trained estimator can transfer to real world well. Utilizing this, we can train control policies to predict reconstructed high-quality 3D motion flow, which is then translated to robot actions in downstream tasks for control. We evaluate our proposed components and system in several real world object manipulation and tool-use tasks. Our system reduces 3D motion field estimation error by over 50% compared to latest works and achieves ∼55% zero-shot success rate on average where prior approaches fall short ($\lesssim 10\%$). Remarkably, we also show that the policy trained solely on human (hand) videos is capable of performing fine-grained manipulations, such as precise insertions – a level of skill not previously shown in this setting.

In summary, our main contributions are as follows. 1). We propose to use object-centric 3D motion field for robot learning from videos and present a novel learning framework for extracting this representation for control. 2). We present a simple and novel architecture that can learn to see and predict object-centric 3D motion field in the real world for control. With this, we can teach robots new skills with human videos as the **only** training data. 3). We validate our proposed components in the real world. We demonstrate that our motion extraction pipeline can significantly reduce motion estimation error by over 50%. For its robotics application, we significantly outperformed existing approaches and show that our policies trained solely on human videos can achieve finegrained manipulation skill for the first time, demonstrating the potential of video-based robot learning.

## 2    Preliminaries

### 2.1    What Can Be Learned from Videos?

We begin by discussing what knowledge can be learned from videos and why we focus on extracting and learning object movements. Recently, some works propose to extract low-level human finger movement from videos and aim at directly retargeting them to robots [51, 36, 32, 20, 21]. Although this might be a feasible approach if robot embodiment is highly similar to human hand, we argue that learning direct embodiment control is both unnecessary and challenging. On one hand, the ultimate knowledge we want to learn is how each object in the view should move in each task, and it does not matter too much how we control a robot to generate such object movement as long as we can achieve it. The state-of-the-art robots have built-in functionality to realize arbitrary object movements: i.e., by calling well-established foundational grasping policy and task-space SE(3) movement commands, and therefore we should focus on extracting object motion. On the other hand, generating reliable action through naive retargeting is hard [57, 24]. In some cases, the movement performed by a human hand might not be realizable by a robot (e.g. when robot has different or fewer fingers) and action retargeting will be undefined. Actually, existing methods typically require the human hand to act as a gripper and avoid in-hand manipulations in the training videos. This is not natural for human at work and introduces extra burdens. We conclude with the following assumption and proposal.

**Assumption 1**    We assume we have access to a reliable robot grasping/ungrasping (i.e. object attaching/detaching) policy. This is a reasonable assumption given success in previous work on robot grippers [11] or hands [61, 56]. Different robots should have their specific grasping policy.

> **Proposal**    We propose to focus on extracting and learning object 3D motions for robot learning from videos. Combining an object flow prediction model with the assumed grasping policies above, a robot can solve a wide range of tasks.

### 2.2    Necessity of Depth Perception

To extract the object 3D motion from video effectively, we assume RGBD videos as training data. While there is growing interest in leveraging RGB-only information for action extraction, we argue that incorporating depth information is necessary for accurate estimation and learning. Otherwise, there exist infinitely many 3D transformations to realize observed pixel motion. Even if we assume some priors over underlying 3D movement, this is still extremely challenging and brittle. For example, suppose that we have a point $(X, 0, Z)$ in the camera frame and it is translating along $z$-axis. Our goal is to recover this small $z$-axis motion $\Delta Z$ from pixels. Assuming a pinhole camera with focal length $f$, the observed pixel movement on $x$-axis is $\Delta x_p = \frac{fX}{Z+\Delta Z} - \frac{fX}{Z} \approx -\frac{fX}{Z^2}\Delta Z$, and we

---

However, we may require some "functional knowledge" as constraints: e.g. do not hold the sides of a cup/ should grasp the tool handle. Therefore in the long run, besides object motion, we also need to consider extracting contact (semantical affordance).

rearrange it as $\Delta Z = -\frac{Z^2}{fX}\Delta x_p$. The problem is that $\Delta x_p$ might be noisy in practice and it can lead to huge estimation error in $\Delta Z$ due to the large slope $-Z^2/fX$, especially near $X = 0$ (i.e. when the camera is looking at the object), and even a 1-pixel error can result in over 1 centimeter difference! This error is disastrous for robot manipulation task requiring high accuracy – to overcome this we need robust subpixel-level trackers. Although this might be possible in the future, having RGBD video instead of RGB video can make learning much easier. Therefore:

**Assumption 2**   We require access to RGBD videos collected through pinhole cameras with known camera intrinsics (i.e. camera focal length $f_x$, $f_y$ and center $c_x$, $c_y$), instead of general RGB videos collected by unknown camera with distortions. Although it might seem like a waste not to reuse RGB videos already available on the internet, accumulating new RGBD videos can be easier than one may think of – Million hours of videos are being produced every day [58] and many existing daily camera devices are already equipped with depth sensing (e.g. latest mobile phones and wearable devices).

## 2.3   3D Motion Field

As we have discussed, our goal is to extract an object-centric 3D motion field from RGBD videos for control. In this paper, a 3D motion field [42, 23, 40] is a dense, image-based representation defined as follows.

**Definition (3D Motion Field)**   Given a pair of $H \times W$ images $I_0$, $I_1$ (current frame and next frame), the 3D motion field $F$ of $I_0$ is defined as 4-channel image tensor in $\mathbb{R}^{H \times W \times 4}$. The first channel $F[:,:,0]$, denoted as $F_{depth}$, is the depth value of each pixel in $I_0$. The remaining three channels $F[:,:,1:4]$, denoted as $F_{motion}$, represent the underlying 3D movement of each pixel between the 2 frames under the $I_0$ camera frame.

Note that we include depth in this definition. Then, given the camera intrinsics and the 3D motion field, we can reconstruct the position and movement of every pixel in the 3D space.

# 3   Phase I: Seeing 3D Motion Field in Noise

Our first step is to extract accurate 3D motion fields from noisy RGBD videos. We first discuss a very simple pipeline for this purpose as suggested by latest works [59] and its fundamental limitations, and then we introduce our improved approach. The discussion below assumes the depth observation of two consecutive images $I_0$ and $I_1$, and a dense pixel correspondence computed by a video tracker.

## 3.1   Direct Approach

We first assigns the camera depth to the $F_{depth} = F[:,:,0]$ depth channel. Then, we run a pixel tracker (e.g. Cotracker [17]) to decide the pixel correspondence between $I_0$ and $I_1$. Let a pixel $(x, y)$ in $I_0$ correspond to $(x', y')$ in $I_1$. Since we have camera intrinsics and their depth values $Z$ and $Z'$, we apply camera inverse projection to get their 3D coordinates $(X, Y, Z)$ and $(X', Y', Z')$, and set $F_{motion} = F[y, x, 1:4] = (X', Y', Z') - (X, Y, Z)$, i.e., the 3D space movement. The procedure above assumes a static camera but it also applies to moving camera (transform $(X', Y', Z')$ to $I_0$ coordinate frame first).

The direct method can be effective if we have perfect depth and perfect tracker, but unfortunately, this is untrue in practice. First, the commonly used depth cameras are sensitive to lighting and particularly erroneous when it comes to moving objects – there are numerous holes (missing values) and white noises across the depth image. Second, even if we might have high quality learning-based binocular depth sensing method to improve the depth accuracy, noises from the pixel tracker can also lead to significant errors in motion. For instance, if the pixel tracker mistakenly maps a foreground object pixel in $I_0$ to a background pixel or a hidden pixel in the next frame $I_1$ (which is common for object boundary pixels), we will obtain incorrect depth $Z'$ and consequently wrong $X'$ and $Y'$ as $X' = Z'x'/f_x$ and $Y' = Z'y'/f_y$. In both cases, we will obtain a wrong motion field. Due to this, prior works typically use intensive heuristics-based filtering to reduce outliers in observations. Nevertheless, heuristics-based filtering can be unreliable, hence not eliminating the problem thoroughly. This will result in a noisy motion field representation and make the subsequent prediction a hard problem.

## 3.2   Our Improved Approach: Learning to See 3D Motion Field

To remedy this issue, we propose to learn a 3D motion field estimator to reconstruct the groundtruth smooth 3D motion field with the noisy sensor measurements. As shown in Figure 2 and 4, the

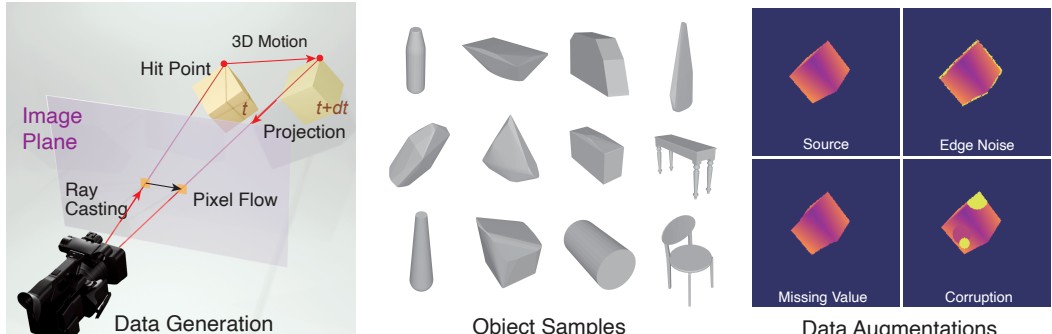

Figure 3: (Left) Phase I Synthetic Data Generation. We randomly generate object and 3D motions, and use ray casting and projection to obtain 3D pixel flow input and the 3D motion field label. (Middle) Random Objects for Data Generation. (Right) Depth Samples and Data Augmentations.

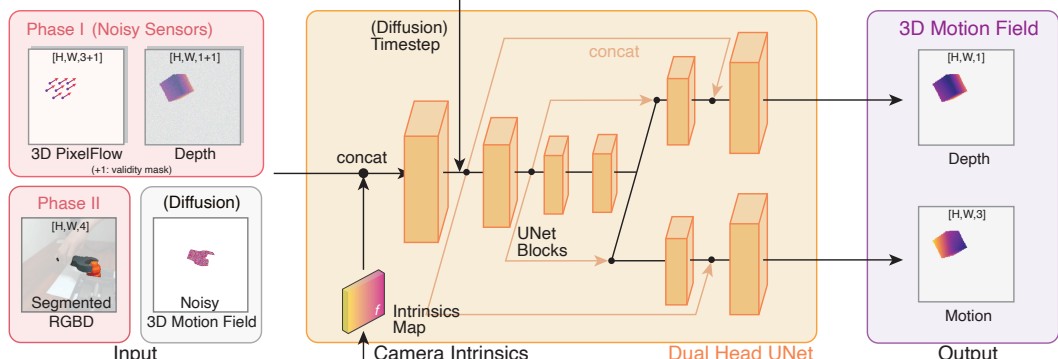

Figure 4: Model Architecture. The most important design is a dense intrinsics map feature concatenated to the input, which contains key information for reconstructing the groundtruth 3D flow. Note that Phase I and II train separate models: Phase I uses noisy object depth and 3D pixel flow as input and Phase II uses segmented RGBD and a noisy 3D motion field (in the case of Diffusion Model). For Phase I, we also use a validity mask value channel to indicate which feature values are defined.

main inputs to this model are noisy dense object 3D pixel flow (2D pixel flow ($x$, $y$-axis) and depth flow ($z$-axis)) and the noisy depth image. The output is the reconstructed 3D motion field. Since all the input and output information are geometrical (no RGB textures), we propose to use a simulator to generate training data due to minimal sim-to-real gap for geometrical data. The dataset, model, and training method are as follows.

**Dataset Generation**   We use the objects in the ShapeNet dataset [5] and some randomly generated regular rigid bodies as training objects. We rescale all the objects to regular daily object sizes, which is about 4~20cm on each dimension. Then, we randomly generate a camera with a random field of view (FoV) between 40~55° (common camera FoV), and randomly place the object in the camera view to render the initial depth frame. Then, we generate a random twist motion $\mathcal{S} = [v, w]$ to move (translate and rotate) this object for several steps. We calculate both the 3D pixel movement and the groundtruth 3D motion of each observed pixel in the initial frame through ray casting followed by transformations as shown in Figure 3. We generate 8M samples at $256 \times 256$ resolution for training, which can be produced with 1 NVIDIA L40 GPU in less than 12 hours.

**Data Augmentation**   During training, we use diverse data augmentations to simulate the noise effect of each sensor observations, and the underlying idea is relevant to the Denoising Autoencoder [43] which reconstructs signals from sensor noises for downstream processing. The most common noise for depth is random missing values, white noises, and wrong values (especially for moving objects), and we apply these effects on the depth image. For the pixel flow input, since we find existing trackers are usually off by a few pixels, we apply random Gaussian noise as augmentation. Besides, we also apply random dropout on the pixel flow input, i.e., only using part of the flow vectors for prediction. This also allows us to use partial, sparse pixel flow for inferring 3D motion field, so that we can speed

up the data labeling process. Finally, we also apply subset masking to the input feature map. With this, we can approximate complex object contours with simple objects.

**Model: 3D Motion Field Estimator**   We use a dual head UNet [33] model as our 3D motion field estimator $f$. This model predicts $F_{depth}$ and $F_{motion}$ through two separate low-level decoder branches $f_{depth}$ and $f_{motion}$. This is to reduce interference near the output as these predicted values have different semantical meanings. Besides, we also append a dense, "intrinsic" map feature $I_{map} \in \mathbb{R}^{H \times W \times 4}$ to the image, whose elements are given by

$$I_{map}[y, x] = ((y - c_y)/f_y, (x - c_x)/f_x, 1/f_y, 1/f_x). \tag{1}$$

$I_{map}$ contains crucial low-level information for accurate $F_{motion}$ prediction. To understand this, let us consider motion prediction along $x$-axis as an example. Since $X = (x - c_x)Z/f_x$, by taking differential we have $dX = (Z/f_x)dx + [(x - c_x)/f_x]dZ$. The (noisy) 2D pixel motion $dx$, depth motion $dZ$, and depth $Z$ are already included as input, while the remaining $1/f_x$, $(x - c_x)/f_x$ terms cannot be inferred by a CNN architecture without position embeddings. Hence, we also include them explicitly as input as well for predicting $x$-axis motion $dX$. These features will directly propagate to the last layers in UNet through skip concatenations. We show their importance in Figure 8.

**Training**   We apply a weighted Huber loss ($\| \cdot \|$) as a stable supervision to train this model:

$$\mathcal{L} = \mathbb{E}_{x, F, M \sim \mathcal{D}_{sim}} \| M \odot (f_{depth}(x) - F_{depth}) \| + \alpha \| M \odot (f_{motion}(x) - F_{motion}) \|. \tag{2}$$

In the loss function above, $\mathcal{D}_{sim}$ is the generated synthetic dataset. $M$ is the object mask and $\odot$ is the (broadcast) elementwise product, so the dense motion field prediction loss is only applied on the object part. $\alpha$ is a weighting hyperparameter. We will also present a diffusion-based architecture and objective in the next section, but for this motion estimation task, a direct prediction without further refinement is already good enough. We use the AdamW optimizer [22] to train this model and the training procedure takes about 1 day with 16 NVIDIA A100-40GB GPUs.

**Discussion I: Motion and Geometry Synergy** A key idea in our prediction task is the synergy between the motion and geometry (depth) – the motion clue can be used to recover the missing or wrong depth value. One example is shown in Figure 5. If we observe that some pixels move together with others whose depth values are known, we can reasonably infer their depth — particularly when the observed object is rigid. This formulation can also be considered as masked pretraining of 4D (time plus 3D) geometrical data. Most importantly, the tracks on the 2D plane are usually very visually accurate ($<$ 3 pixel error), making such "masked" reconstruction feasible. In the experiments, we find that adding motion and camera intrinsics information can significantly improve depth prediction accuracy.

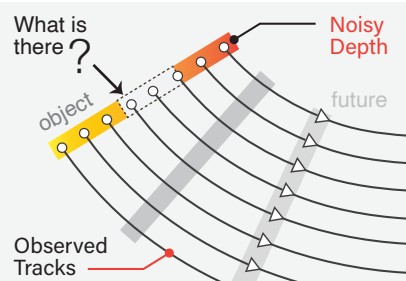

Figure 5: Guess what is there? Object tracks can be used to recover missing or wrong depth values.

**Discussion II: Extension to Moving Camera**   In our current implementation, we assumed a static camera. However, since the data is produced in simulation, this framework can be extended to estimating 3D motions in videos captured by moving cameras (through simulating camera movement). Then again, one can append an additional dimension of (noisy) camera motion as input to the estimator and use it to process general videos (the camera motion in videos can be inferred via monocular simultaneous localization and mapping (SLAM) [10]). We leave this extension as a future work.

## 4   Phase II: Predicting Object 3D Motion Field for Control

Now that we have a model to see the groundtruth 3D motions from noisy sensors, we are ready to build our control policies with human videos.

**Dataset: Human Videos**   We only require human object interaction video dataset $\mathcal{D}_{human}$ to train our control policy. We first process the dataset through our learned estimator and existing foundation models. Specifically, we first apply foundation segmentation model SAM2 [31] in video mode to extract the segmentation of all the task-relevant objects in each frame. Then, we use pixel trackers (CoTracker3 [17]) to extract the noisy 3D pixel flow of each object point and lift them to an

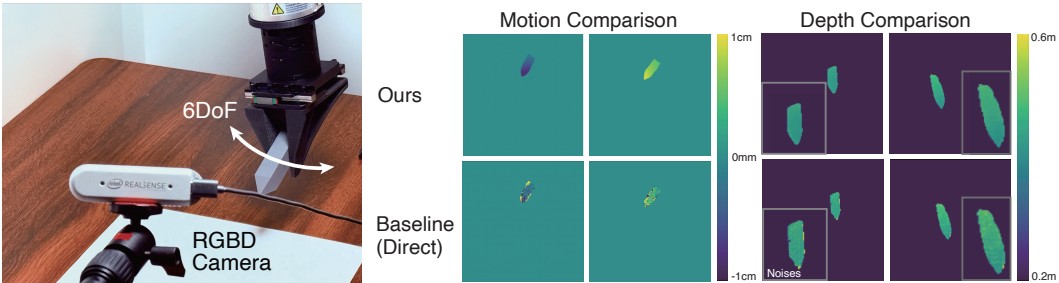

Figure 6: (Left) Experimental Setup. (Right) Qualitative Results on "Pen" (Left Figure). Our method produces smoother motion field and depth compared to baseline. This is essential for accurate control.

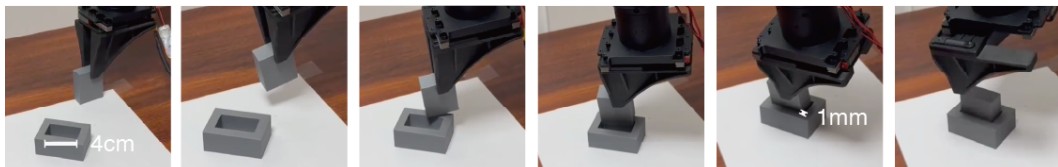

Figure 7: A rollout of fine-grained insertion. Our method can achieve high precision, even if we are observing the motion from 40cm away without a wrist camera. However, we observe an adjustment behavior resembling bang-bang control – it is still challenging for the policy to insert accurately in one-shot as human. Fortunately, it still captures rough directions for adjustment to finish the task.

accurate 3D motion field through our pretrained estimator. However, an underlying requirement is that we require the object consistently visible (i.e. not fully occluded) throughout the video segments, so we have to discard the fully occluded segments and learn from all the remaining partially occluded and fully visible segments. We leave the full occlusion case to future investigation.

**Model and Training**    Then, we train a policy network $\pi$ to predict these labeled 3D motion field with the segmented RGBD image as input. Since our motion field is image-shaped, we can use either a Gaussian or a diffusion model (policy) [14, 38, 7] for its accurate prediction. We reuse most of the dual-head UNet architecture as our policy $\pi$. We apply the following general regression objective for training (for both diffusion and Gaussian policy):

$$\mathcal{L}_\pi = \mathbb{E}_{o,F,M \sim \mathcal{D}_{human}} \|M \odot (\pi_{depth}(o, \tilde{F}, t) - F_{depth})\| + \alpha \|M \odot (\pi_{motion}(o, \tilde{F}, t) - F_{motion})\|. \tag{3}$$

In this objective above, $o$ is the segmented RGBD image observation, $F$ is the groundtruth object 3D motion field (desired action over the object) labeled by our estimator, and $M$ is the mask of the corresponding object extracted by SAM2. $(\tilde{F}, t)$ is the noised motion field sample and timestep and only apply to diffusion model. As our policy network only uses task-relevant object information as input and does not observe any embodiment-specific information, the gap between human domain and robot domain is minimal. However, we still find it important to apply a random masking data augmentation to objects. Besides, for diffusion model we also find it useful to use "masked noise sample" as input – this simplifies training and makes the model more robust.

**Deployment**    In the inference time, we need to convert the predicted 3D motion field $F$ to the robot action. As the object is already firmly grasped by the robot, this conversion is straightforward. For each pixel on the object mask, given $F$, we compute their current and future 3D coordinates in the camera frame through camera inverse projections, resulting in two point clouds $P_0, P_1 \in \mathbb{R}^{N \times 3}$. Importantly, these two point clouds have point-wise correspondence (as opposed to unordered point clouds that require ICP [2]) so that we can directly solve a SE(3) transformation $\mathbf{T}_o = \{\mathbf{R}, \mathbf{t}\}$ for aligning them. We minimize $\|\mathbf{R}P_0^T + \mathbf{t} - P_1^T\|^2$, which has a closed form solution (Kabsch method [16]). As there are outliers in $P_0$ and $P_1$ inevitably, so we also use RANSAC [13] to improve the quality of $\mathbf{T}_o$. Then, suppose the camera pose in robot base frame $\{b\}$ is $\mathbf{T}_{bc}$, our desired robot action can be computed as $\mathbf{T}_a = \mathbf{T}_{bc} \mathbf{T}_o \mathbf{T}_{bc}^{-1}$. The conversion from object 3D motion field to SE3 action is very fast at around 300-1000Hz, introducing minimal overhead to control loop.

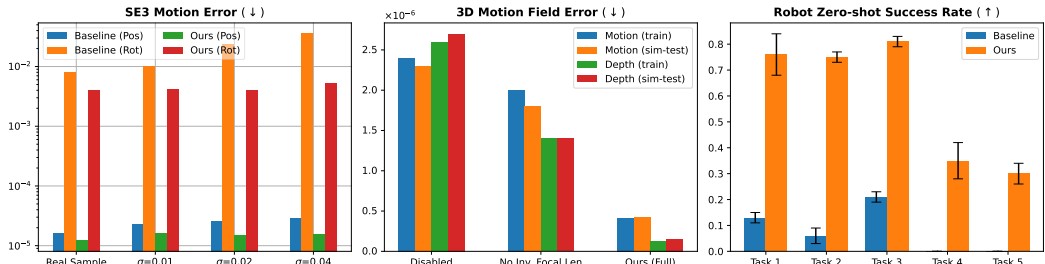

Figure 8: (Left) SE3 motion estimation performance in real world. Our method achieves lower error compared to baseline. (Middle) Intrinsics Map Ablation Studies. Both inverse focal length and coordinate map are crucial. (Right) Real world Task Success Rate (3 seeds). Baseline is the latest General Flow [59]. Other recent methods fail on our setup due to their limitations (Table 2).

## 5 Experiments

In this section, we demonstrate the effectiveness of our 3D motion field estimator and our control policy through real world experiments.

**System Setup**    We use a widely-used Intel D435 RGBD camera at $640 \times 480$ resolution for video dataset collection at 30Hz (Figure 6). During training, we crop the image to $480 \times 480$ and rescale it to $256 \times 256$. To get a high-quality raw depth, we use the native temporal and spatial filters provided by the camera SDK. We use an XArm7 robot arm with a parallel-jaw gripper for the test dataset collection and robot experiments. We did not use a wrist camera in the experiments as previous works. In the experiments, the manipulated object is typically 40~50cm away from the camera.

### 5.1    Evaluating 3D Motion Field Estimator

**Setup**    We capture RGBD video of randomly moving objects in the real world with their corresponding ground truth transformations as the test set. We program the robot arm to grasp random objects and wave them in front of the camera. Since the robot gripper grasps the object, we can directly obtain the groundtruth object 3D transformation by reading and calculating robot gripper pose transformation. We use objects of diverse aspect ratios and apply diverse robot motions.

**Main Results**    We show the results of our method and the latest baseline "direct" method in Figure 8. We report the MSE of recovered object translation and the $F$-norm of object rotation matrix error. Note that we use the gripper frame as the base for representing these transformations. We find that our approach has significantly lower error compared to the baseline. We also visualize some examples in Figure 6. Our method successfully reproduces smooth depth and motion field even if the input depth is very noisy. We refer readers to appendix for more details.

**Adversarial Robustness**    We test robustness further through adversarial attack in real world experiments by injecting Gaussian noise of different intensities into the depth observation (which then propagates into the computed 3D pixel flow input). We find that the baseline performance deteriorates quickly, while the error of our method remains at the same level. However, this is somewhat expected since the same kind of noise is also applied during our training process.

**Ablation Studies**    We also ablate intrinsic maps as shown in Figure 8 (Middle). We find that both coordinates and *inverse focal length* are crucial for successful learning, confirming our derivation. Surprisingly, the focal length value plays a critical role in motion prediction even for a relatively small FoV variation around 10 degrees.

### 5.2    Robot Learning from Videos

**Real world Tasks**    In this section, we test if our method can acquire object manipulation skills from human videos. We introduce the following tasks to benchmark the performance:

1. *Pick, Rotate, and Place*. The robot is required to pick up an object and/or rotate it and put it at a target location. This task is considered as successful only if the object is in correct pose in the end.

2. *Line Tracking*. The robot is required to pick up a pen-like flashlight and control it to track a cable on the table. This task is considered successful if the robot can finish the tracking trajectory with spotlight focusing on the cable in the process. The procedure is monitored by a human.

Table 2: Technical Feature Comparisons. Our method is free from many limitations of existing works. Note that this list is non-exhaustive and we refer readers to the text for more discussion.

| Method | UniPi [9]'23 | ATM [49]'24 | Track2Act [3]'24 | GFlow [59]'24 | Im2Flow2Act [52]'24 | SPOT [15]'24 | TI [6]'25 | Ours |
|---|---|---|---|---|---|---|---|---|
| No Pose Estimation | ✓ | ✓ | ✓ | ✓ | ✓ | × | × | ✓ |
| No Robot Data (Zero-shot) | × | × | × | ✓ | × | ✓ | ✓ | ✓ |
| Close-loop *Motion* Control | ✓ | ✓ | ✓ | ✓ | × | ✓ | ✓ | ✓ |
| Depth Robustness | N/A | N/A | N/A | × | N/A | N/A | N/A | ✓ |
| Distractor Generalization | × | × | × | × | × | × | × | ✓ |

3. *Tool Use I: Pushing*. The robot is required to pick up a tool and push one object to a goal.

4. *Tool Use II: Wrench*. In this task, the robot is required to pick up a wrench and use it to tighten a nut by a round. This task can be viewed as a more challenging version of pick, rotate, and place since the process is kinematically constrained and requires precision.

5. *Insertion*. In this task, the robot is required to pick, rotate, and insert an item into a slot (hole). There is only 2.5mm tolerance so this requires very high precision.

We collect around 50-150 human videos for each of these tasks. The data collection procedure for each of the tasks lasts around 2-15 minutes, depending on task complexity. During the evaluation, we ensure that the grasping and the object segmentation is correct for each of the evaluated method (otherwise we restart that trial).

**Main Results**   We show the success rate of different methods in Figure 8 Right. We find that our method significantly outperformed the other evaluated methods. During the deployment, the baseline method will quickly deviate from the correct moving direction/trajectory. Our method not only solves the tasks but also follows the human-demonstrated path accurately throughout the evaluation (see Appendix). We attribute its success to accurate and smooth motion estimation. Besides this, we also observe the expected robustness to the background variations during experiments due to the use of object-centric input representation.

**Ablation Studies**   We also study the design choices of our policy architecture and training. We find that for fine-grained tasks, it is important to apply a diffusion model even if the human has tried to act as consistently as possible. Compared to the Gaussian policy head, the diffusion policy can produce high-quality, accurate motion fields which is important for success. Besides, we also find it important to mask out unnecessary regions in the diffusion model reverse step. Otherwise, the irrelevant noise in non-object regions can slow down training

Table 1: Policy Learning Ablation for Fine-grained Tasks.

| Setting | Success |
|---|---|
| w/o Diffusion (Diff.) | 0.0% |
| w/o Diff. Masking. | 0.0% |
| w/o Masking Data Aug. | 5.0% |
| Full | **35.0%** |

and harm performance. Finally, we find it important to apply object masking augmentation during training, as the object's silhouette under the robot gripper differs from that under a human hand, which leads to a subtle domain gap.

## 6   Related Works

**Robot Learning from Videos**   We summarize the most relevant studies in Table 2. A key aspect of existing methods is how they represent actions. Some approaches rely on direct video frame prediction [12, 9, 54, 4] for control while recent research explores compact and informative representations, such as 2D pixel flow [49, 41, 52, 3], point cloud flow [35, 59], and 3D poses [15, 6]. While these approaches offer certain advantages, each has notable limitations, as previously discussed. Among them, point cloud flow is most relevant to our approach. However, we distinguish our work by employing an image-based 3D motion representation with a denoising architecture to refine the extracted 3D information. Another related line of research involves embodiment-specific action retargeting [51, 32, 20], which can be infeasible in general setups. In contrast, our embodiment-agnostic approach aims to extract common motion knowledge for control.

In addition to action representation, another challenge is domain alignment—ensuring that the learned model can effectively transfer to the robot domain. Recent works tackle this by employing ego

masking or inpainting [20, 18]. We have a similar idea but adopt a more object-centric approach by using task-relevant objects as input. While not entirely new [62], we are the first to apply this concept within the context of learning from human video demonstrations. In a broader context, human videos have also been used as auxiliary data source to pretrain vision-language-action models/policies [**?**, 37] or define RL tasks [29, 30, 28] in simulation, which are orthogonal directions.

**3D Vision and Video Analysis**  Our method is related to 3D reconstruction from videos [10, 26, 34]. Some works in this area have utilized 3D representations such as Neural Radiance Fields (NeRF) [25], Gaussian Splatting [19], and Pointmaps [47, 60, 46] for scene *geometry* reconstruction, while our work focuses on *motion* reconstruction. In recent 3D motion extraction research, some works propose both optimization-based and end-to-end tracking method [27, 50, 45]. Our method shares a similar goal but differs significantly in focus and underlying assumptions. These works typically focus on long-range (time) tracking and assume the raw depth as the reference even if it can be noisy and inaccurate. In contrast, we focus on recovering depth and precise instantaneous motion from noisy temporal depth sensing. In this sense, our work is more aligned with depth estimation [53, 48] — areas traditionally explored within computational photography for 3D. While many existing approaches aim to recover depth from monocular or stereo RGB images as well, our method takes a distinct direction by leveraging the interplay between depth and motion to improve 3D understanding. Finally, we note that our approach is compatible with the methods above – it can be integrated with existing depth and track estimation methods by using their depth and track outputs as inputs.

## 7    Conclusion

In this paper, we have demonstrated a novel framework for learning robot control policies from human videos using object-centric 3D motion field representations. Our approach overcomes key limitations of existing representations by introducing a robust 3D motion estimator and a dense flow prediction architecture, enabling better cross-embodiment transfer and background generalization. Experiments demonstrate substantial improvements in motion estimation and success rates across diverse real-world tasks compared to prior works, and the unprecedented effectiveness in handling precise manipulation tasks. Our approach opens up new possibilities for leveraging scalable human video datasets to train versatile and generalizable robotic agents.

### Acknowledgement

This work is part of the Google-BAIR Commons project. The authors gratefully acknowledge Google for providing computational resources. Zhao-Heng Yin is supported by the ONR MURI grant N00014-22-1-2773 at UC Berkeley. Pieter Abbeel holds concurrent appointments as a Professor at UC Berkeley and as an Amazon Scholar. This research was conducted at UC Berkeley and is not affiliated with Amazon.

### Broader Impact

Our method targets one of the most significant challenges in robot learning: data. By addressing this issue through human video data, we open the door to scaling up data for the development of foundational robotic agents.

### Limitations

We identify the following limitations in this work as directions for future research. First, we did not consider knowledge extraction in the case of full occlusion – we may need separate action representations for mining action knowledge from fully occluded data. Second, we mainly consider grippers and it would be interesting to study the case of robotic hands – for which we may need to train motion-conditioned control policy rather than solving an optimization problem as we did. Third, in this study, we assumed a fixed camera although we have shown a potential way to scale it to moving cameras (Section 4 Discussion II). Finally, it would be interesting to handle the case of (very) soft-bodies like cloth – for which we should not extract an SE(3) transformation from the whole object movement, but instead only consider a subset of motion field around the contact point. In summary, there are several directions for extending this work and making it a useful industry-level solution.

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
