# OpenReview forum: "Object-centric 3D Motion Field for Robot Learning from Human Videos"
_NeurIPS.cc/2025/Conference — NeurIPS 2025 spotlight_

### Official Review · Reviewer_xCja · 2025-06-23

**Clarity:** 3
**Significance:** 2
**Originality:** 2
**Rating:** 4
**Confidence:** 3

**Summary:**

### Summary

This paper introduces a novel approach for robot learning from human videos, leveraging an object-centric 3D motion field to represent actions. The core of their framework is a 3D Motion Field Estimator, which is initially trained using simulated data to refine noisy motion fields. This estimator is then applied to improve the quality of human manipulation data, which is subsequently used to train a robot policy.

**Questions:**

Broadly speaking, the core idea of this paper—predicting the **point cloud flow (or motion)** of manipulated objects and then converting that into robot control signals—bears a strong resemblance to prior work like Track2Act and GFlow.

While the paper undeniably presents a more refined workflow, incorporating elements such as dataset augmentation, deployment-time optimization, and leveraging SAM2 for object-centric understanding, it feels more like a highly engineered and detailed implementation of existing concepts rather than a breakthrough in innovation or a source of new insights. For instance, some of the "new" capabilities, such as **distractor generalization**, seem to stem from the integration of SAM2, and **depth robustness** appears to be a benefit of introducing the **3D Motion Estimator**.

Therefore, I'd like the authors to elaborate further on the **innovative aspects and core insights** of your proposed method. Specifically:

* Are there any capabilities that **only the proposed method can achieve** and that previous methods simply cannot?
* Or is the proposed method primarily a **better-designed, more deployment-ready version** of existing approaches?

I'm hoping for a discussion that highlights the unique contributions and fundamental advancements that set this work apart, beyond just improved engineering and integration of other state-of-the-art components.

**Ethical Concerns:**

["NO or VERY MINOR ethics concerns only"]

**Final Justification:**

In my initial and current assessment, I found that the proposed method bears a resemblance to previous approaches, and some of its novel capabilities appear to stem from integrating existing models.

However, it's undeniable that the **workflow is meticulously designed**, eliminating the need for manual dataset processing and enabling scalability. Insights from other reviewers and the authors' rebuttal have shifted my perspective; the distinction from previous methods now seems less critical. The proposed method successfully enables video-based policy learning to achieve fine-grained operational tasks, a capability that many advanced VLA models currently lack.

Therefore, I agree that this work contributes significantly to the community, and the paper provides valuable insights into pipeline design.

**Limitations:**

Please see Questions

**Quality:**

3

**Strengths And Weaknesses:**

### Strengths

* The paper is well-structured, making it easy to follow with a clear and logical flow.
* The proposed method is simple, direct, and easy to understand. Its workflow demonstrates a rigorous and effective engineering design.

---
### Weaknesses

* The paper's method bears a resemblance to previous approaches, and some of its novel capabilities appear to stem from integrating existing models. I'll elaborate on this specific point in the "Questions" section.

---

> ### Author Rebuttal · Authors · 2025-07-31
>
> Thank you for the thoughtful review! We would like to highlight several key novel contributions of our work:
>
> **1. More challenging data setups:**
> Unlike existing systems such as Track2Act, which require in-domain robot data, our method operates solely on human demonstration videos. In contrast to methods like GFlow, which rely on extensive point cloud filtering and heuristic, hand-engineered preprocessing (see their appendix), our approach avoids such manual steps by leveraging a learned motion denoiser. A detailed comparison with other systems can be found in Table 2. We made strictly less assumptions over data in this paper.
>
> **2. Scalability:**
> Many prior works, such as ATM and General Flow, use point-based prediction architecture, which makes it difficult to integrate them with standard generative modeling frameworks. Our method uses an image-based motion representation that aligns naturally with image generative models, facilitating greater scalability.
>
> **3. More precise tasks:**
> Previous works have mostly focused on coarse manipulation tasks (e.g., mostly pick-and-place motion of large objects) and have not explored fine-grained scenarios involving small objects or tool use. Our paper demonstrates that such challenging tasks can, in fact, be addressed using a purely video-based learning pipeline.
>
> We acknowledge that, at a high level, our method may resemble existing works in predicting "flow"-like representations for control. However, we argue that the specific choice of representation and the way to extract it are critical factors that determine the system’s capability. In summary, our contributions are twofold: 1. We propose a scalable pipeline that relies on significantly fewer data assumptions than existing methods. 2. We show that this pipeline can handle fine-grained manipulation tasks that prior approaches struggle with.
>
>
> Together, these results provide the first strong evidence that simple, pure video-based policy learning is a viable and promising direction for solving complex, high-precision tasks—and now we are confident in its potential to scale further.

---

> > ### Comment · Reviewer_xCja · 2025-08-01
> > **Reply to the rebuttal**
> >
> > Thank you for your detailed response. I have re-examined the paper's details based on your explanations. While I still find the work bears a strong resemblance to previous methods, I agree that many of your design choices significantly enhance the workflow's efficiency. The proposed method enables video-based policy learning to perform fine-grained operational tasks effectively.
> >
> > Therefore, I am persuaded by your arguments and will increase my rating accordingly. (As you may notice that NeurIPS 2025 seems to intentionally prevent reviewers' updated ratings from being visible to authors until final decisions are made, so I am unsure if it is appropriate to state the exact rating here.)
> >
> > Thank you again for your rebuttal.

---

> > > ### Author Response · Authors · 2025-08-01
> > > **Thank you for increasing the score and for the review!**
> > >
> > > Thank you for raising the score and for  the insightful review. We’ll continue working in this exciting direction!

---

### Official Review · Reviewer_gWxp · 2025-07-01

**Clarity:** 4
**Significance:** 4
**Originality:** 3
**Rating:** 5
**Confidence:** 4

**Summary:**

The paper demonstrates that the motion field is key to learning from human videos for robotic policy learning. It first analyzes the drawbacks of current action representation and points out that the proposed object-centric motion field has clear advantages. Then, it argues that the learning objective from the human video is object 3D motions and demonstrates the necessity of using RGBD video. After that, it raises the challenges when learning 3D motions from videos, which is the accumulated noise. It finally designs the 3D motion field estimator to reduce noise, which makes it possible to train video policy models from the accurate motion fields. It also discusses the deployment issues. Experiments on the real-world setups demonstrate the effectiveness of the proposed method.

**Questions:**

The paper is well-written and acceptable. The motivation is clear, the proposed method is effective, and the discussion of each part of the proposed model is adequate. I only have a few concerns (see above).

**Ethical Concerns:**

["NO or VERY MINOR ethics concerns only"]

**Final Justification:**

I keep my original rating.

**Limitations:**

Yes

**Quality:**

4

**Strengths And Weaknesses:**

## Strength

- The writing is quite clear and fluent. The motivation of the proposed method is well established and discussed.
- The proposed model that learns motion field from video is interesting and has clear advantages compared to previous learning strategies.
- Experiments show that the proposed learning strategy performs well in real-world scenarios.

## Weakness

- In Phase 2, the model only takes object information as input, which is pre-processed by SAM2. Is it suitable for real-world settings? We may always have to manually instruct SAM2 to obtain the segmentation of the target object.

- I found that when training the policy model in Phrase 2, it only uses RGBD image to infer the 3D motion field. This confuses me because object motion in one image can correspond to multiple potential motion fields. How can the predicted motion field be ensured is the desired motion field? For example, given a bottle image, the policy model could predict arbitrary motion trajectories of that bottle. It seems that it cannot control the predicted motions. Do I miss something?

---

> ### Author Rebuttal · Authors · 2025-07-31
>
> Thanks for the review! Our responses to your questions are as below.
>
> Q1. Thank you for the question. In most of the scenarios, we may want to prompt the robot with language during deployment. There are many existing vision-language models (VLMs) and open-vocabulary object detectors (e.g. Grounding-DINOv2) capable of grounding target objects in human language instructions to a box/point in the current scene automatically. We believe that this will not be an issue in the near future due to rapid advances in VLM and vision research. In our setup, the object only needs to be labeled once at the beginning. Its mask is then robustly tracked throughout execution using SAM2. We have experimented with Grounding-DINOv2 + GPT and it works well in our setups.
>
> Q2. Thank you for the question. The human videos we use are task-relevant—for example, if the task involves picking up a bottle and pouring it, the predicted motion field consistently reflects that intention (e.g., the bottle moves upward and then tilts).
> We acknowledge that motion in human demonstrations can be multi-modal—for instance, the bottle might be lifted slowly in one video and quickly in another. Our method addresses this by using an image-based generative model that learns to produce plausible motion trajectories observed in the training data. In other words, it can generate either a slow or fast lifting motion based on the current visual context, but the motion itself always remains valid and task-relevant.
> Furthermore, the predicted 3D object motion field defines how each visible point on the object should move in 3D space, relative to the camera frame. This makes the motion representation unambiguous—unlike 2D motion fields, which can be affected by depth ambiguities or projection artifacts.

---

> > ### Comment · Reviewer_gWxp · 2025-08-02
> >
> > I sincerely thank the authors for their rebuttal. It addresses my concerns. I keep my original rating.

---

### Official Review · Reviewer_e4mh · 2025-07-01

**Clarity:** 4
**Significance:** 3
**Originality:** 3
**Rating:** 5
**Confidence:** 3

**Summary:**

This paper introduces an object-centric 3D motion field as the action representation for learning robot control policies from human videos. The motion field is defined as a 4-channel image, where the first channel is depth and the next three channels are 3D movements of each pixel from the current image frame to the next. The paper proposes a two-phase learning framework. In phase 1, a denoising 3D motion field estimator (with extra intrinsic map features as input) is pretrained in simulated environments. In phase 2, object instances are segmented (SAM2) from human videos. A policy network is then trained on lifted (CoTracker3) and denoised (phase 1) 3D motion field extracted from human videos. The proposed 3D motion field-based policy is evaluated on 5 real-world scenarios, in which *Tool Use II: Wrench* and *Insertion* are difficult manipulation tasks. The experiment results demonstrate that the proposed method succeeds even when no robot demonstrations are provided, and the evaluation performance is better than the baseline.

**Questions:**

1. (Weakness 1) How to address the 3D motion field difference (the pixels' 3D movement part) caused by the speed difference between humans and robots?
2. (Weakness 2) What is the exact frame interval used to compute the 3D movement used in the extraction of 3D motion field?
3. (Weakness 3) Is the depth channel duplicated in the phase 2 model's input? If yes, why is this redundancy introduced?

**Ethical Concerns:**

["NO or VERY MINOR ethics concerns only"]

**Final Justification:**

The authors' response has addressed my concerns, including how to handle the gap between human demonstration and robot deployment, the frame interval for the extracted motion, and the specific form of the representation. The rating is kept as Accept.

**Limitations:**

Yes.

**Paper Formatting Concerns:**

None.

**Quality:**

3

**Strengths And Weaknesses:**

Strengths:
1. This paper proposes 3D motion field. This representation is aligned to observation frames (unlike pointcloud tracking methods) and can be integrated with input image observations.
2. This paper introduces the intrinsic map as an extra feature input to the model, and the authors explain the motivation and insights for introducing it.
3. This paper is clearly written.

Weaknesses:
1. (Motion field movement difference) The proposed motion field covers the 3D movement of each pixel between the current frame and the next frame. However, the amount of the movement is determined by the motion speed of the human/robot between the frames. Typically, humans move the object at a much faster speed, leading to a larger movement between the frames. The paper does not provide the explanation of how the 3D Motion Field addresses the differences in 3D motion fields arising from the speed differences between humans and robots when performing tasks.
2. (Motion field frame interval) In addition to 1., the exact value of the frame interval between the current frame and the next frame used in the extraction of 3D motion fields is not directly provided. There is information on the fps of the camera used in deployment, but it is unclear whether that fps aligns with 3D motion field extraction.
3. (Duplicated input channel) In phase 2, the input contains both the 3D motion field and the segmented RGBD. As the 3D motion field already contains the depth information, the depth channel in the input seems to be duplicated. The paper provides no explanation or analysis of this potentially redundant input format.

---

> ### Author Rebuttal · Authors · 2025-07-31
>
> Thanks for the review! Our responses to your questions are as below.
>
> Q1. Thank you for the question. Our method directly generates the next target positions for all object points, which are then used to compute the corresponding robot poses. The robot proceeds to the next prediction cycle only after successfully reaching the current target. We can command the robot to the target within the desired time interval.
>
> Q2. We use a control frequency of 10 Hz in our experiments, corresponding to a 0.1-second interval between predictions.
>
> Q3. In the second phase, the depth in the camera input is still very noisy, and our policy can output a cleaner depth map for motion recovery.

---

> > ### Comment · Reviewer_e4mh · 2025-08-06
> >
> > The authors' response has addressed my concerns.

---

### Official Review · Reviewer_tpQo · 2025-07-03

**Clarity:** 3
**Significance:** 3
**Originality:** 3
**Rating:** 4
**Confidence:** 3

**Summary:**

This paper proposes an object-centric pipeline for learning robot policies from human videos. One contribution of this paper is a fine object 3D motion estimator, and the other contribution is the prediction algorithm of object-centric 3D motion (the scene after the target is manipulated). The algorithm mostly operates in 3D, with compact usage of classic techniques such as ICP and RANSAC. Overall, this is a good work. Please see my comments below.

**Questions:**

please see my comments above.

**Ethical Concerns:**

["NO or VERY MINOR ethics concerns only"]

**Limitations:**

yes

**Paper Formatting Concerns:**

no such issues.

**Quality:**

3

**Strengths And Weaknesses:**

1. As the authors do not opensource the model, can the human video lists be available? Otherwise the results are not reproducible.

2. The paper does not mention how the manipulator is controlled. Through inverse kinematics?

3. Did the authors test the algorithm in complex background scenes? The paper says that the sensor is mounted 40cm~50cm away from the target. Therefore, are the figures in Fig.7 cropped images? not the raw data?

4. I assume that the hole structure shown in Fig.7 is fixed on the table.

---

> ### Author Rebuttal · Authors · 2025-07-31
>
> Thank you for appreciating our work! Our responses to your questions are as below.
>
> **Q1.** Thank you for the suggestion. We will release the synthetic data generator, pretrained model, and data labeling pipeline to ensure reproducibility and enable researchers to apply our method in their own task setups.
>
> **Q2.** Yes, the manipulator is controlled via inverse kinematics after computing the next target pose.
>
> **Q3.** The image in Figure 7 was recorded from a different viewpoint solely for visualization purposes and does not reflect the policy's actual observation. In practice, the object appears significantly smaller in the policy's visual input.
>
> **Q4.** Due to the difficulty of the precise insertion task, we opted not to rigidly fix the base structure. This simplification helps avoid damage that can occur when the stiff robot controller applies excessive force. However, it's important to note that this is still very hard as our control policy is purely vision-based (from 40-50cm away) without any force or tactile information, and accurately inferring the insertion angle from vision alone is inherently challenging (and this may result in a slight base movement). For the success trials, the base may be moved by around 1-2mm based on our measurement on the base corners. We hypothesize that in the future, the use of higher-resolution depth cameras or multi-view sensing setups (e.g. fusing the 3D motion flow prediction) could help improve performance in such tasks.

---

### Decision · Program_Chairs · 2025-09-17

**Decision:**

Accept (spotlight)

**Comment:**

The paper proposes object-centric 3d motion fields as a new action representation for robot learning from human videos, addressing limitations of pixel flow and point-cloud flow. A denoising motion field estimator is introduced to robustly reconstruct fine 3D object motions from noisy depth, along with a dense prediction architecture that supports cross-embodiment transfer and background 3d. Policies trained on these motion fields achieve over 50% lower estimation error and ~55% zero-shot success rates in diverse real-world tasks, including fine-grained manipulations such as insertion, where prior methods fail. This establishes a scalable framework for learning robot skills directly from human video data.

In the original round of reviews the committee appreciated the clear motivation and strong real-world validation of the proposed framework (R#gWxp, R#e4mh), the simple yet effective engg. design and workflow (R=xCja,tpQo), and the clarity and logical presentation of the method (reviewers gWxp and e4mh). The overall technical solidity and potential impact on video-based robot learning was also recognized.

The rebuttal effectively clarified key technical questions -- human–robot speed mismatch and frame intervals (reviewer e4mh), SAM2 reliance and motion ambiguity (reviewer gWxp), and novelty beyond prior flow-based methods (R=xCja) -- with reviewers confirming their concerns were resolved. R#e4mh kept their accept rating after being fully satisfied; gWxp reaffirmed their positive assessment; reviewer #xCja explicitly raised their score, acknowledging that the clarified design choices strengthen novelty and enable fine-grained video-based policy learning. The AC concurs with the unanimous positive consensus of hte committee’s reviews and post-rebuttal discussions.